# The Role of the Aryl Hydrocarbon Receptor (AhR) and Its Ligands in Breast Cancer

**DOI:** 10.3390/cancers14225574

**Published:** 2022-11-14

**Authors:** Stephen Safe, Lei Zhang

**Affiliations:** Department of Veterinary Physiology and Pharmacology, College of Veterinary Medicine and Biomedical Sciences, Texas A&M University, College Station, TX 77843, USA

**Keywords:** AhR, breast cancer, agonist, ligand, TCDD

## Abstract

**Simple Summary:**

The aryl hydrocarbon receptor (AhR) is expressed in breast cancer cells and tumors and in some studies, the AhR is a negative prognostic factor for patient survival. Structurally diverse AhR ligands have been extensively investigated as anticancer agents in breast cancer cells and tumors and show efficacy in both estrogen receptor (ER)-positive and ER -negative breast cancer cells. Moreover, synthetic AhR ligands are being developed and have been in clinical trials for treating breast cancer. In contrast, other reports show that AhR ligands enhance mammary carcinogenesis and in a few studies opposite results are observed for the same AhR ligands in comparable breast cancer cells lines. This paper attempts to provide an extensive, unbiased review of the contrasting effects of AhR ligands in breast cancer and points out that future research will be required to resolve these conflicting results.

**Abstract:**

Breast cancer is a complex disease which is defined by numerous cellular and molecular markers that can be used to develop more targeted and successful therapies. The aryl hydrocarbon receptor (AhR) is overexpressed in many breast tumor sub-types, including estrogen receptor -positive (ER+) tumors; however, the prognostic value of the AhR for breast cancer patient survival is not consistent between studies. Moreover, the functional role of the AhR in various breast cancer cell lines is also variable and exhibits both tumor promoter- and tumor suppressor- like activity and the AhR is expressed in both ER-positive and ER-negative cells/tumors. There is strong evidence demonstrating inhibitory AhR-Rα crosstalk where various AhR ligands induce ER degradation. It has also been reported that different structural classes of AhR ligands, including halogenated aromatics, polynuclear aromatics, synthetic drugs and other pharmaceuticals, health promoting phytochemical-derived natural products and endogenous AhR-active compounds inhibit one or more of breast cancer cell proliferation, survival, migration/invasion, and metastasis. AhR–dependent mechanisms for the inhibition of breast cancer by AhR agonists are variable and include the downregulation of multiple genes/gene products such as CXCR4, MMPs, CXCL12, SOX4 and the modulation of microRNA levels. Some AhR ligands, such as aminoflavone, have been investigated in clinical trials for their anticancer activity against breast cancer. In contrast, several publications have reported that AhR agonists and antagonists enhance and inhibit mammary carcinogenesis, respectively, and differences between the anticancer activities of AhR agonists in breast cancer may be due in part to cell context and ligand structure. However, there are reports showing that the same AhR ligand in the same breast cancer cell line gives opposite results. These differences need to be resolved in order to further develop and take advantage of promising agents that inhibit mammary carcinogenesis by targeting the AhR.

## 1. Introduction

The aryl hydrocarbon receptor (AhR) is a basic helix-loop-helix protein that binds the environmental toxicant 2,3,7,8-tetrachlorodibenzo-p-dioxin (TCDD) with high affinity and mediates the toxic and biologic effects induced by this compound and structurally-related halogenated aromatics [1,2,3,4,5]. The classical mechanism of AhR-mediated gene expression and functions involves the ligand-dependent formation of the AhR and the AhR nuclear translocator (ARNT) protein as a heterodimer, which in turn binds cognate cis elements in target gene promoters [6]. The cis elements or xenobiotic response elements (XREs) contain a core GCGTG pentanucleotide sequence and variable flanking nucleotides [6,7,8]. This classical mechanism of action of the AHR:ARNT complex which targets sequence-specific cis-elements is similar to that described for many members of the nuclear receptor (NR) superfamily of intracellular receptors such as estrogen receptors (ERs, ESR1) [9,10,11].

Among all intracellular receptors, the AhR is the only receptor identified in molecular toxicology studies focused on determining the mechanism of action of TCDD and structurally related compounds [1,2]. The discovery of the AhR as a “toxicant” receptor has subsequently been a significant hindrance in development and clinical applications of AhR ligands for the treatment of multiple diseases. Overcoming the concerns regarding the potential toxicity of AhR ligands was due to several factors, including the development of AhR knockout (AhRKO) mouse models [12,13,14] and discoveries showing that many AhR ligands are “health promoting” compounds [15,16,17,18,19]. Differences in AhR ligand persistence may be related to their dioxin-like toxicities. In this review article, the role of the AhR and its ligands as inhibitors of breast cancer in cellular and in vivo models will be investigated. Several studies support a role for AhR ligands as inhibitors of breast cancer and this includes some studies in this laboratory on TCDD and related compounds as antiestrogens associated with ligand-dependent inhibitory AhR-ERα crosstalk [20,21,22]. There is extensive support from cell culture and in vivo studies indicating that the AhR is a target for breast cancer therapy [23] and human clinical trials using the AhR ligand “aminoflavone” have been carried out for treating breast and other cancers [24,25,26,27]. Moreover, studies from several laboratories show that many structural classes of AhR ligands also inhibit some aspects of mammary carcinogenesis [23]. In contrast, there are also reports showing that AhR ligands enhance breast cancer growth and development [28,29,30,31] and there are other examples of AhR/AhR ligands exhibiting both tumor suppressive and tumor promoter-like activities for specific cancers [32,33,34,35,36]. Some of these differences are irreconcilable. However, in breast cancer there are several factors that can contribute differences in the role of the AhR/AhR ligands in breast cancer and this includes the following:breast cancer cell context which includes differential expression of the ER and other as yet unidentified factors,breast cancer complexity associated with multiple classifications of tumors based on differences in their histopathology, gene expression, and other clinical parameters,ligand structure and the fact that selective AhR modulators (SAhRMs) exhibit tissue/cell-specific AhR agonist or antagonist activity,other mechanisms of action of the AhR which involve altered genomic and non-genomic (e.g., cell membrane) pathways that may be differentially be affected by AhR ligands some of which also activate more than one receptor. An example of dual receptor ligands are the polyaromatic hydrocarbons (PAHs) and other compounds which bind both the AhR and ER [36,37,38,39].

## 2. Selective AhR Modulators (SAhRMs)

Initial studies on the AhR and the steroid hormone NRs identified exogenous (e.g.,: TCDD) or endogenous (e.g., steroid hormones) ligands that act primarily as receptor agonists. However, for the AhR and other nuclear receptors, it was soon recognized that many different structural classes of ligands also bound the receptor and could act as tissue/cell-and even gene specific agonists, antagonists or partial agonists/antagonists. For example, the AhR binds structurally diverse industrial and synthetic compounds, PAHs, pharmaceuticals, mycotoxins, multiple classes of health promoting phytochemicals including flavonoids, polyphenolics, heteroaromatics such as indole-3-carbinol (I3C), microbial metabolites, and 1,4-dihyroxy-2-naphthoic acid (DHNA) [15,16,17,18,19] (Figure 1 and Figure 2). In addition, some endogenous compounds, including 6-formyl (3,2-b) carbazole (FICZ), 2-(1′-H-indole-3-carbonyl) thiazole-4-carboxylic acid methyl ester (ITE), tryptophan metabolites such as kynurenine and other gut microbial products, and leukotrienes may play a role as endogenous ligands for the AhR [40,41,42,43,44,45].

The selectivity of AhR ligands in terms of their tissue-specific agonist or antagonist activity has been reported in breast and other cancer cell lines and also has been recently reviewed [15]. It is striking that the RNAseq analysis of TCDD and related toxicants are highly variable with respect to their differentially expressed genes. A landmark study of 596 drug-related compounds identified a sub-set of 147 compounds that were evaluated in several Ah-responsive assays, including receptor binding, reporter gene activation and CYP1A1 gene expression [53]. They observed multiple differences among these pharmaceutical compounds to activate putative Ah-responsive endpoints. For example, 59% (81/137) of the compounds induced hepatic CYP1A1 mRNA in mice but did not bind or activate the AhR in vitro. Only nine of these compounds exhibited both in vitro and in vivo activity as AhR agonists in the complete panel of assays which included cytochrome P450 induction in mouse cancer cell lines and liver (in vivo). The selectivity of these AhR-active pharmaceuticals has been further investigated in our laboratory in breast and pancreatic cancer cells [54,55,56].

The pharmaceutical-derived AhR agonists identified in the screening study [54], including flutamide, leflunomide, mexiletine hydrochloride, nimodipine, omeprazole, sulindac and tranilast were investigated in breast cancer cells. Their activity as inducers of CYP1A1 and CYP1B1 in MDA-MB-468 and BT474 breast cancer cells was structure, response and cell type-specific. Compared to TCDD, induction of CYP1A1 was more robust in BT474 than MDA-MB-468 cells, whereas for most of these compounds the reverse was true for the induction of CYP1B1. 4-Hydroxytamoxifen induced minimal (<25% of maximal induction observed for 10 nM TCDD) CYP1A1 and CYP1B1 expression in both cell lines and with the exception of induction of CYP1B1 (50% of maximal induction observed for 10 nM TCDD), tranilast was also a weak inducer of CYP1A1 and CYP1B1 [54]. The pattern of CYP1A1/CYP1B1 induction in MDA-MB-231 cells also differed from that observed in MDA-MB-468 and BT474 cells. The selectivity was also observed for the effects of these pharmaceuticals on the invasion (Boyden chamber) of MDA-MB-231 cells. TCDD (minimal), omeprazole and tranilast inhibited invasion at sub-toxic concentrations, whereas no effects were observed for 4-hydroxytamoxifen, flutamide, leflunomide, mexiletine, nimodipine, and sulindac [55]. Using omeprazole as a model, it was also shown that inhibition of MDA-MB-231 cell invasion by omeprazole was reversed by AhR antagonists and AhR knockdown (siAhR) and inhibition of invasion was primarily due to AhR-dependent downregulation of CXCR4, which was observed both in vitro and in vivo [55]. These highly variable ligand-dependent results in breast cancer cells were observed for pharmaceuticals that were all AhR-active in liver and liver cancer cells, demonstrating that these compounds are SAhRMs. Moreover, there is evidence for SAhRM-like activity for many other structural classes of AhR ligands [15,16].

## 3. AhR in Breast Cancer: Prognostic Significance

There are multiple genes/gene products expressed in breast cancer and other tumors that can predict overall survival or recurrence of disease and they also may be useful for selecting appropriate treatment regimens [57]. These markers, which include receptors, may or may not be indicative of their functional activity or predict effects of therapeutic regimens. There were some differences in the nuclear and extranuclear distribution of AhR protein in non-pathological breast ductal epithelial cells and invasive ductal carcinoma and AhR overexpression was associated with better prognosis of ER-negative and ER-positive invasive ductal carcinoma patients [58]. In another study on 436 breast cancer cases, it was concluded “that AhR expression is not a prognostic factor in breast cancer” [59]. There were correlations between AhR, and levels of several genes associated with inflammation and high levels of AhR repressor (AhRR) mRNA which predicted enhanced patient survival. This might suggest that since AhRR inhibits AhR function due to competition for ARNT, then high AhR levels would be negative prognostic factors; however, this was not observed in a prospective study of 1116 patients where correlations between multiple prognostic factors and their combination with AhR expression were evaluated for their prognostic value. Low cytosolic AhR levels and positive aromatase were associated with more aggressive ER negative (ER^-^) tumors; however, AhR tumor genotypes did not correlate with AhR protein levels [60]. Another report indicated that the predictive value of the AhR was dependent on the lymph node status of the patient and concluded “that AhR is a marker of poor prognosis for patients with LN-negative luminal-like BCs” [61]. The results suggest that the prognostic value of AhR levels (mRNA and protein), intracellular location and AhR polymorphisms with respect to patient survival/disease recurrence is complex and dependent on many other factors, including prior patient treatment protocols [58,59,60,61].

## 4. Role of the AhR and Its Ligands as Inhibitors Breast Cancer in Cellular and Rodent Models

(i) **Long-term feeding and Huggins model:** Knockdown of the AhR in mice results in lesions in multiple tissues and immune function abnormalities [62,63,64,65]. However, the loss of this receptor and development of murine mammary tumorigenesis has not been reported. Studies on the potential effects of TCDD on the development of mammary cancer in rodent models were initially investigated as part of the risk assessment of TCDD, and there is also evidence for impacts of this toxicant on mammary gland development [66,67,68]. A long-term feeding study in female Sprague Dawley rats showed that TCDD decreased benign mammary tumor formation in one study [69] but this response was minimal in another chronic toxicity feeding study [70]. Another breast cancer model involves 7,12-dimethylbenz[a]anthracene (DMBA)-induced mammary cancer in female Sprague Dawley rats which can then be subsequently treated to identify potential antitumor agents. This model developed by Huggins and coworkers [52] depends on the administration of DMBA to 50 day-old rats and appropriate metabolic activation of DMBA which produces a maximal mammary tumor response. The prenatal administration of TCDD altered mammary gland development [66,67,68] and enhanced DMBA-induced mammary cancer formation, whereas AhR activation during pregnancy decreased DMBA-induced tumor promotion [71,72]. The Huggins protocol has been used to determine the anticancer activities of AhR ligands. The results show that several AhR agonists, including TCDD, 3,3^1,^4,4^1^-tetrachlorobiphenyl (TCBP), diindolylmethane (DIM) and substituted DIM analogs, and 6-methyl-1,3,8-trichlorodibenzofuran (MCDF) inhibit the formation and growth of mammary tumors [46,47,48,49,50,51]. These studies are consistent with the antiestrogenic effects of AhR ligands, resulting in antitumorigenic activity.

(ii) **AhR Function:** In many tumor types, the AhR alone exhibits tumor promoter or tumor suppressor like activity (Table 1) [73,74,75,76,77,78,79,80,81,82,83,84,85,86,87,88,89], and this can be observed in animals or cells after knockdown or overexpression studies [33,34]. Although the loss of AhR in mice does not affect mammary tumorigenesis, studies in breast cancer cell lines give variable results. For example, knockdown of the AhR by RNA interference increased or did not affect proliferation in BT474 and MDA-MB-468 cells, respectively [73]. In MCF-7 breast cancer cells in a mouse xenograft model, AhR expression was not required for mammary tumorigenesis [74]. In another study, the loss of AhR did not affect the growth of MCF-7 cells; however, TCDD inhibited the growth of AhR-expressing and AhR-KO cells [75]. Another report showed that overexpression of the AhR enhanced MCF-7 cell growth [76,77]. AhR knockdown in ER-negative, MDA-MB-231 cells decreased proliferation and wound healing but induced apoptosis and inhibited tumor growth in an athymic nude mouse xenograft model. [78]. In contrast, knockdown of the AhR in MDA-MB-231 and SKBR3 cells increased invasion [79], whereas other studies gave variable results [55,80]. Stable AhRKO MDA-MB-231 cells were analyzed in a 3-dimensional microfluidic invasion assay that examined both functional and genomic differences with respect to loss of the AhR. MDA-MB-231-AhRKO cells exhibited enhanced invasion characteristics and transient AhR expression in these cells’ decreased invasion, confirming that the AhR inhibited invasion. In contrast, the loss of AhR in these cells decreased proliferation and proliferation-related genes, indicating that the receptor played a role in cell proliferation, which was in contrast to its effect as an inhibitor of invasion [81]. Thus, the role of the AhR alone in breast cancer is variable and is breast cancer cell type-dependent; current evidence favors a pro-oncogenic phenotype for the AhR, but this needs to be further investigated in multiple breast cancer cell lines. Nevertheless, the expression of the AhR in breast cancer and in breast cancer cell lines offers the opportunity for investigating the potential for AhR ligands as chemotherapeutic agents for treating breast cancer.

(iii) **Synthetic halogenated aromatic AhR ligands:** Confirmation that the AhR is a target for developing anticancer drugs for treating breast cancer has been extensively investigated using different structural classes of AhR ligands and an array of breast cancer cell lines, including ER-positive (T47D, MCF-7 and ZR-75), ER-negative (MDA-MB-231, MDA-MB-468, MDA-MB-436, MDA-MB-157, MDA-MB435, BT26, CRL2335, BT20, BT549, BT479, HS5787, HCC38, mouse 4T1) and HER-2/ErbB2 positive (BT474, SKBR3 and MDA-MB-453) cells. AhR-active ligands have primarily been used in these studies, but it is also possible that some of these compounds also target other receptors and proteins. AhR specificity is confirmed in many studies by results from AhR knockdown (KO) or cotreating with AhR antagonists. Many of these studies investigate the effects of AhR ligands on one or more of cell proliferation/cell cycle progression, survival/apoptosis invasion/migration, metastasis, inflammation, changes in mRNA and microRNA expression, and as SAhRMs most AhR ligands selectively modulate responses.

Table 1 summarizes the effects of TCDD and structurally related halogenated aromatics as AhR agonists in breast cancer cells (Figure 1). One study used 5 different ligands and 6 different breast cancer cell lines to show that these AhR ligands inhibited breast cancer cell growth, and this was supported by limited KO studies in which loss of the AhR resulted in loss of AhR ligand -dependent growth inhibition [73]. Moreover, it was also shown that TCDD inhibited tumor growth in an athymic nude mouse xenograft model bearing MDA-MB-468 cells [83]. Several other studies reported that TCDD inhibited growth or invasiveness, decreasing pro-invasion genes (CXCR4 and MMP-9) in breast cancer cells [83,84,85,86] and 2,3,7,8-tetrachlorodibenzofuran (TCDF) also inhibited invasion of MDA-MB-431 and T47D cells [86] and BT474 cells [87]. Another early study using MCF-7 cells showed that TCDD inhibited early tumor growth, but this effect was lost during the later stages of the experiment [88]. Wang and coworkers used a syngeneic immune competent mouse xenograft study with mouse mammary cancer 4T-1 cells and showed that TCDD inhibited lung metastasis and metastasis from the primary tumor site but did not affect growth of the primary tumor [89]. In addition, TCDD and DIM suppressed metastasis by targeting SOX4 via microRNAs [86]. Thus, results obtained using TCDD and related AhR ligands indicate that these compounds inhibit some pro-oncogenic functions of breast cancer.

**(iv) Synthetic AhR ligands including pharmaceuticals (Figure 2)**: Initial studies in this laboratory identified MCDF as a partial AhR antagonist which inhibited induction of CYP1A1 by TCDD in cell culture, whereas MCDF exhibited AhR-dependent antiestrogenic activity in the mouse uterus and breast cancer cells. Table 2 summarizes the inhibitory effects of MCDF on the growth and invasion of breast cancer cells [73,87] and this compound also inhibited growth of tumors in an athymic nude mouse model bearing MDA-MB-231 cells [87]. Synthetic aminoflavone [5-amino-2-(4-amino-3-fluorophenyl)-6,8-difluoro-7-methyl-4H-1-benzopyran-4-one; NSC-688228] and its prodrug conjugate are AhR ligands that have been in clinical trials for breast cancer chemotherapy [24,25,26,27,90,91,92,93,94]. In contrast to most other AhR ligands, aminoflavone and some related synthetic aromatic amines require AhR-dependent activation of CYP1A1 and other drug metabolizing enzymes. The induced enzymes result in metabolic activation of the pro-drug, which causes downstream cellular damage and pathways leading to cell killing. Two aminobenzothiazole derivatives, namely 2-(4-amino-3-methlyphenyl)benzothiazole (DF-203) and 2-(4-amino-3-methlyphenyl)-5 fluorobenzothiazole (SF-203) are AhR ligands that have also been in clinic trials for treating breast cancer [95,96,97,98,99,100,101,102,103]. These compounds are similar to aminoflavones, undergo metabolic activation and induce cytotoxic downstream pathways, including oxidative stress. Some ER-negative cell lines which exhibit low CYP induction were relatively insensitive to the cytotoxic effects of the aminobenzothiazoles. Several other AhR ligands, including a novel naphthylamide (2-(2-aminophenyl) -H-benzo [d,e]isquinoline-1,3 [2H]-dione) (NAP6) and related compounds [104,105] and N,2-dimethyl-N-[1,2-dimethylindol-5-yl] quinazoline-5-amine (^#^12), [106] also inhibit breast cancer cell/tumor growth. Both AhR ligands may act in part via metabolic activation and #12 also inhibits microtubule polymerization. (Z)-2 (3,4-dichlorophenyl)-3-(1H -pyrrol-2-yl) acrylonitrile (ANI-7) and related compounds are AhR ligands that exhibit antitumor activity in breast cancer and there is some evidence that metabolic activation also contributes to their effects [107,108,109]. CGS-15943 is an aminoglycoside identified in a screen for inhibitors of multidrug resistance plasmid [110] that was also identified as an AhR ligand that exhibits anticancer activity [111]. This compound induced apoptosis in MDA-MB-468 cells and its activity as a SAhRM was confirmed primarily in liver cancer cell lines. A recent study also showed that carbidopa, a drug used in treating Parkinson’s disease, inhibited breast cancer cell and tumor growth through the AhR-dependent degradation of ER [112], and this pathway has also been observed for TCDD [20].

Hu and coworkers [54] investigated 596 pharmaceuticals for their AhR activity and identified only 9 compounds that were active in vitro in mouse liver and in mouse hepatoma cells. As indicated above, some of these AhR active compounds were subsequently screened for their activity in breast cancer cells. Among the AhR -active (liver) pharmaceuticals 4-hydroxytamoxifen mexiletine, flutamide, leflunomide, nimodipine omeprazole, sulindac and tranilast, all but 4-hydroxytamoxifen and mexiletine inhibited migration of MDA-MB-468 cells [55]. In contrast, only nimodipine and omeprazole inhibited MDA-MB-231 cell invasion, and for omeprazole, this response was reversed after knockdown of AhR [87]. Tranilast has also been investigated in mouse 4T1 cancer cells and inhibits cell growth and invasion and migration, as well as tumor growth and metastasis in a syngeneic mouse model [113]. Tranilast was also anticarcinogenic in BT474 and MDA-MB-231 human breast cancer cell lines [114,115]. Beta-Naphthoflavone is a well-known “non-toxic” AhR ligand which inhibits MCF-7 but not MDA-MB-231 cell proliferation, cell cycle progression and related genes [116]. Both raloxifene and 4-hydroxytamoxifen are two antiestrogens that are also AhR ligands that inhibit apoptosis and differentiation of breast cancer cells [117,118] and they are part of a group of compounds that are dual AhR-ER and ligands. Results summarized in Table 2 demonstrate that structurally diverse ligands inhibit mammary carcinogenesis in multiple breast cancer cell lines and in vivo xenograft models. The specific responses observed are ligand structure- and cell context-dependent and this includes compounds such as aminoflavone that have been in clinical trials for breast cancer and other cancers. A role for the AhR in mediating these responses has been confirmed in cell cultures and in vivo studies; however, contributions from the drug acting on its traditional target cannot be excluded.

(v) **Natural products and endogenous AhR ligands**: Natural products such as the polyphenolics are also AhR ligands; however, many of these compounds bind multiple receptors or have other activities which contribute to their anticancer activities in breast and other cancers (Figure 3). 1, 1-Bis (3^1^-indolyl) methane (DIM), the dimeric metabolite of indole-3-carbinol (I3C) binds the AhR and several studies confirm the activity of this compound as an inhibitor of breast cancer cell and tumor growth [48,85,119,120,121,122]. DIM inhibits growth of ER^+^ and ER^+^ cancer cell lines and inhibits growth of carcinogen induced mammary tumors in both orthotopic and syngeneic mouse models. These effects have been associated with CXCR4/CXCL12 downregulation and induction of miR-212/132 [86]. The antiestrogenic activity of DIM was also reported in a human nutritional intervention study with healthy women that express BRCA [122]. I3C also binds the AhR with lower affinity than DIM and I3C inhibits breast cancer cell growth and migration [123,124,125,126]; some of this activity may be due to the facile conversion of I3C into DIM. Indolo-[3,2b}-carbazole is another AhR-active metabolite of I3C that inhibits breast cancer cell migration [125]. Although many flavonoids have been characterized as AhR ligands [127,128], very few have been investigated for their effects on breast cancer. Luteolin was particularly effective against MD-MB-231 cells [129] and the prenylflavone icaritin exhibited dose-dependent antiestrogenic activity but also inhibited the growth of MCF-7 cells in culture and in a xenograft model in vivo [130]. Icaritin also downregulated ER expression and this was presumed to be a major pathway for mediating cell growth inhibition. 3,4,5-Trihydroxy-6-methylphthaldehyde (Flavipin) is a fungal metabolite that inhibits T47D, and MDA-MB-231 cell growth, invasion and migration and these responses are blunted after AhR knockdown [131]. Glyceollins (Figure 3) are soybean phytoalexins that are AhR ligands and both glyceollin I and glyceollin II inhibit migration of MDA-MB-231 cells [132]. Camalexin, an indole phytoalexin, 2-hydroxy-6-tridecylbenzoic acid and the polyphenolic gallic acid, are also phytochemical AhR ligands that exhibit anticancer activity in breast cancer [133,134]. Gallic acid inhibits tumor growth in athymic mouse xenograft models bearing MDA-MB-231 and T47D cells, and also inhibited growth, migration, and invasion in cell culture [135].

FICZ and ITE are endogenous AhR ligands that may play a role in AhR function and both compounds are inhibitory in breast cancer cells [136,137]. ITE inhibits growth, migration, and invasion of MDA-MB-231 but not MCF-7 cells, and this may be related in part to decreased JAG1 and NOTCH signaling. In contrast, the antiproliferative and antimigration effects of FICZ in MCF-7 cells are associated with several miRs. The tryptophan metabolites indoxyl sulfate and indole propionic acid are AhR ligands and inhibited 4T1 cell and tumor growth (syngeneic mouse model) and EMT and induced oxidative stress [138,139]. The results observed with the natural products and potential endogenous AhR ligands clearly show that these compounds exhibit anticancer activity in breast cancer cells (Figure 4). However, it is also apparent that this activity is response and cell context dependent, which is typically observed for SAhRMs.

## 5. AhR and AhR Ligands Enhance Mammary Carcinogenesis

Although endogenous expression and function of the AhR in breast cancer cells is variable, the effects of structurally diverse AhR ligands (Table 1, Table 2 and Table 3) are primarily associated with selective inhibition of pro-oncogenic responses. This was observed in cells in which AhR knockdown exhibited increased or decreased proliferation, survival, or migration/invasion (Figure 5). Studies by Sherr and colleagues contrast with results summarized in Table 1, Table 2 and Table 3. It was initially reported that the AhR and ReIA cooperatively activated cMyc expression in Hs578T cells and thereby enhanced cMyc-dependent tumorigenesis [140]. In a subsequent study in Hs578T cells, it was reported that the constitutive AhR suppressed cMyc expression and was activated by the AhR repressor (AhRR) but not TCDD [30]. These studies, which focused primarily on the role of constitutive AhR, gave variable results; however, subsequent reports show that in ER-negative Hs578T and SUM149 breast cancer cells that AhR ligands enhance tumorigenesis (Table 4) (Figure 5) and contrast with results summarized in Table 1, Table 2 and Table 3. For example, the tryptophan metabolites kynurenine, xanthurenic, acid (XA) FICZ, and benzol(a)pyrene (BaP) enhance SUM149 cell migration and the AhR antagonist CH223191 inhibited these responses for the former two compounds [141]. Moreover, a newly identified AhR antagonist (CB7993113) [142] and CH223191 also inhibited migration in Hs578T and SUM149 cells, and this is supported by a study showing that the AhR antagonist galangin decreased growth promoting genes in Hs578T cells [143]. Similar results were observed in SUM149, Hs578T and MCF-7 cells where the AhR and its agonists were associated with inducing cancer stem cell characteristics [144]. Suspended ER negative BT549, MDA-MB-231 and SUM159 breast cancers cells express higher AhR levels and AhR inhibition or loss decreased pro-oncogenic pathways. AhR agonists DIM and TCDD enhanced migration of Hs578T and SUM149 cells and complementary results were observed in a zebrafish model [31]. TCDD also induced the inflammatory precursor gene COX-2 and in MCF7-cells DIM inhibited the induction by TCDD [145]. In contrast, it has also been reported that the AhR agonist DIM inhibits mammary carcinogenesis (Table 2). In a study on interactions between tryptophan-2.3-dioxygenase (TD02) and AhR signaling [146], it was observed that cells in suspension exhibited an enhanced response that was associated with expression of higher AhR levels. Knockdown of AhR or treatment with CH223191 decreased MDA-MB-231 and BT549 cell growth and colony formation and kynurenine decreased apoptosis in BT479 cells [146]. Knockdown of the AhR in BT549 and MDA-MB-231 cells in suspension induced apoptosis, which has also been observed in MDA-MB-231 cells treated with AhR ligands that inhibit mammary carcinogenesis (Table 1, Table 2 and Table 3), and these differences need to be resolved. BaP was used as an AhR ligand and it was shown that it induced migration of MDA-MB-231 cells and this response was inhibited by CH223191 [147]. This inhibitory interaction between an AhR ligand and an AhR antagonist is expected but the results showing that the AhR ligand BaP induced migration of MDA-MB-231 cells contrast with the effects of other AhR ligand in this and other cells lines, as summarized in Table 1, Table 2 and Table 3.

There are also studies where the AhR and other ligands or cellular factors play a pro-oncogenic role in breast cancer. For example, several phthalates activate the extranuclear AhR in MDA-MB-231 cells via activation of histone deacetylase 6 (HDAC6) and downstream induction of cMyc [148]. A subsequent study by this group focused on other factors involved in phthalate-AhR interactions in MCF-7 and MDA-MB-231 cells and their data suggest some phthalates may directly activate the AhR -dependent metabolic genes and enhance doxorubicin metabolism [149]. A third study [149] showed that TCDD induced migration of MCF7 cells, which also contrasted with results of previous studies with TCDD and other AhR ligands (Table 1, Table 2 and Table 3). Moreover, they observed that mono 2-ethylhexylphthalate (MEHP) induced MCF7 cell migration that was inhibited after co-treatment with TCDD. The phthalate/AhR/AhR ligand interactions gave some conflicting results [148,149,150] and warrant further investigation due to the importance of environmental/dietary exposures to phthalates. The effects of AhR ligands on drug-induced responses, such as apoptosis in breast cancer, have also been investigated in several cancer cell lines, including MDA-MB-231 and SKBR3 cells treated with doxorubicin, lapatinib and paclitaxel [151]. Cotreatment with TCDD decreased drug induced apoptosis, which was partially reversed by the AhR antagonist 3^1^-methoxy-4^1^-nitroflavone. A complementary study [152] showed that the AhR blunted the effects of doxorubicin on cell viability in MDA-MB-231 cells and this was due in part to AhR regulation of aldo-ketoreductase 1C3. The AhRR decreases availability of functional AhR by competing for ARNT, and using transgenic mice overexpressing AhRR, it was shown that AhRR suppresses mammary tumor development and AhR-dependent growth and the inflammatory gene COX2 (±TCDD) [153]. Similar results were observed in MCF-7 and MDA-MB-231 cells treated with etoposide and doxorubicin; both drugs induced the percentage of apoptotic cells which was further enhanced after cotransfection with an AhRR expression plasmid. These results are also consistent with a role for the AhR in blunting the effects of drug induced cytotoxic responses in breast cancer.

There is also evidence that the AhR plays a pro-oncogenic role in other models of breast cancer. One study reported that ROS levels in BRCA1 and basal-like breast cancer correlated with AhR expression and this increased expression of amphiregulin (AREG), a ligand for the epidermal growth factor receptor (EGFR) [154]. The AhR antagonist CH223191 inhibited AREG expression in HCC1937 and MDA-MB-468 cells and synergistically interacted with the kinase inhibitor erlotinib in BT20, MDA-MB-468 and HCC1937 cells but not MDA-MB-231 cells (due to low EGFR expression). These results demonstrating growth inhibition of MDA-MB-468 cells by the AhR antagonist CH223191 contrasts with previous reports showing that TCDD, MCDF, I3C and other AhR angonists inhibit MDA-MB-468 cell growth (Table 1, Table 2 and Table 3). Protein kinase 6 (PTK6) is also overexpressed in many breast tumors and plays a role in lung metastasis and cell motility in triple negative breast cancer cells [155]. Mechanistic studies show a relationship between PTK6, RhOA and the AhR and AhR activities require the PTK6SH2 domain. In SKBR3 cells, it has also been shown that the AhR agonist 3-methylcholanthrene (MC) binds and integrates the AhR and the G protein estrogen receptor (GPER) [156]. This results in induction of CYP1B1 in cancer-associate fibroblasts and SKBR3 cells. Cyclin D1 was also increased and was inhibited not only by CH223191, but also mithramycin (Sp1 inhibitor), G15 (GPER inhibitor) and TMS (CYP1B1) inhibitor. Endogenous growth of SKBR3 cells was not affected by CH223191, suggesting that MC may be acting as a dual AhR/ER ligand [37,39,40,153], and this system needs to be further investigated.

Several studies have also linked expression of the AhR with inflammatory response pathways in breast cancer cells. For example, phorbol ester, (PMA) alone induced interleukin-8 (IL-8) and IL-6 in MCF-7 cells and PMA in combination with TCDD synergistically enhanced Il-6 but not IL-8 mRNA levels [157]. Similar interactions of PMA and interleukin-1β with other AhR ligands enhanced IL-6 levels. Another report showed that heregulin enhanced AhR levels, IL-6 and IL-8 expression and also increased invasion in a HER2 overexpressing breast cancer cell line. Loss of AhR decreased invasion, IL-6 and IL-eight levels [157]. IL-2 plays an important role in the development of T cell exhaustion [159] and this compromises the effects of the CD8 + T cell dependent-immune response to tumors and infection. IL-2 also enhances AhR expression and metabolism of tryptophan to 5-hydroxytryptophan, an AhR ligand which enhances markers of CD8 + T cell exhaustion [159]. Evidence for the IL-2-AhR-5HTP-dependent activation of T cell exhaustion is supported by human and laboratory animal studies, and this pro-oncogenic role for the AhR and 5HTP suggests that targeting the AhR to inhibit IL-2-dependent initiation of CD8 + T cell exhaustion may be feasible for treating breast cancer. It will also be important to resolve potential differences between this and other studies using a syngeneic mouse model and mouse cancer 4T-1 cells. For example, tumors derived from this cell line exhibit T cell exhaustion and two AhR-active tryptophan -related metabolites, indoxyl sulfate and indole propionic acid inhibited tumor growth [138,139].

## 6. Conclusions

In this review, there is strong evidence that in a large number of breast cancer cell lines the AhR alone exhibits both pro-and anti-oncogenic activity or minimal activity based on results of knockdown experiments. In these cell lines and in some animal models, structurally diverse synthetic, pharmaceutical, natural product and endogenous AhR ligands exhibit pro-oncogenic activity and inhibit one or more of cell/tumor proliferation survival migration/invasion and metastasis (Table 1, Table 2 and Table 3) (Figure 4). In some studies, these responses are reversed by AhR knockdown or AhR antagonists. Moreover, many of these ligands have been tested in clinical trials for breast cancer chemotherapy. Not surprisingly, the mechanisms associated with the anticancer activity of these SAhRMs exhibit selectivity and are dependent on ligand structure and cell context, and for some compounds this includes their differential effects in ER-positive and ER negative cell lines. However, results illustrated in Figure 4 summarize a number of the key genes/pathways that have been characterized, and these include CXCR4, CXCL12, MMP-9, SOX4, and several microRNAs.

In contrast, the treatment of some cell lines such as the inflammatory Hs578T and SUM149 cells, AhR ligands inhibit AhR-dependent pro-oncogenic genes and signaling pathways. Moreover, several in vitro and in vivo studies show that the AhR plays a pro-oncogenic and pro-inflammatory role in mammary carcinogenesis; AhR agonists enhance these responses, while AhR antagonists inhibit them, as outlined in Figure 5. However, in some cases, effects of AhR ligands in the same cell line gave opposite responses and these differences need to be resolved. However, it is clear that AhR ligands effect some AhR-dependent genes and pathways to promote mammary cancer (Figure 5), whereas there is also strong evidence that AhR agonists are potential drugs for clinical application in breast cancer therapy (Table 1, Table 2 and Table 3).

It will be important in the future to identify factors that are responsible for these differences in the anticancer activities of AhR ligands (cell context) in order to use AhR-active compounds as “precision” therapeutics for treating breast cancer.

## Figures and Tables

**Figure 1 cancers-14-05574-f001:**
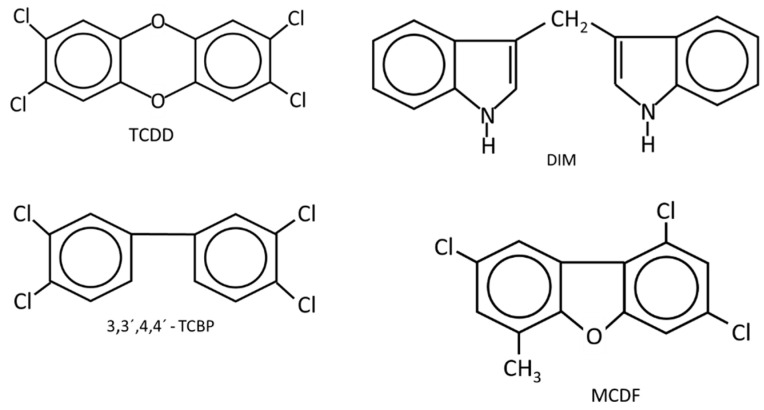
The structures of AhR ligands that inhibit DMBA-induced mammary cancer growth in female Sprague Dawley rats [46,47,48,49,50,51] using the Huggins model [52].

**Figure 2 cancers-14-05574-f002:**
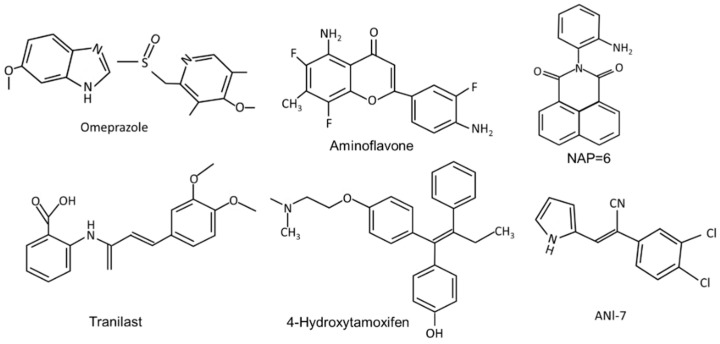
The structures of synthetic compounds, including pharmaceuticals that are AhR ligands.

**Figure 3 cancers-14-05574-f003:**
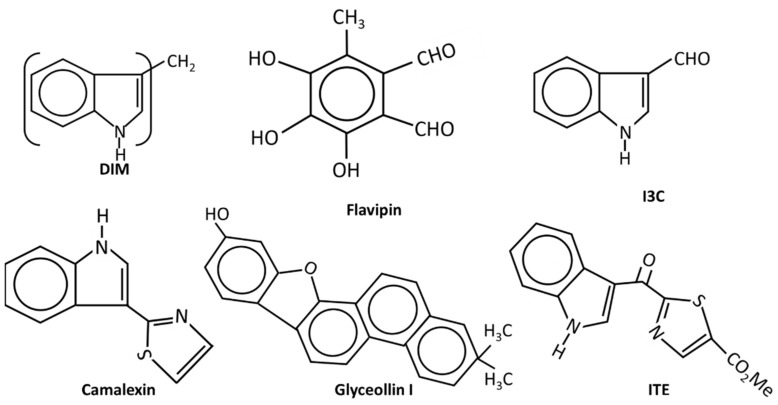
The structures of endogenous and natural product-derived AhR ligands.

**Figure 4 cancers-14-05574-f004:**
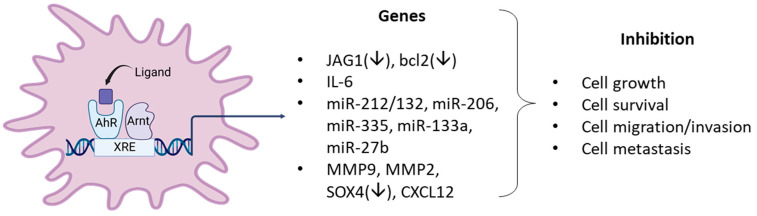
Examples of AhR ligand-activated pathways/genes that result in anticancer activity in breast cancer cells.

**Figure 5 cancers-14-05574-f005:**
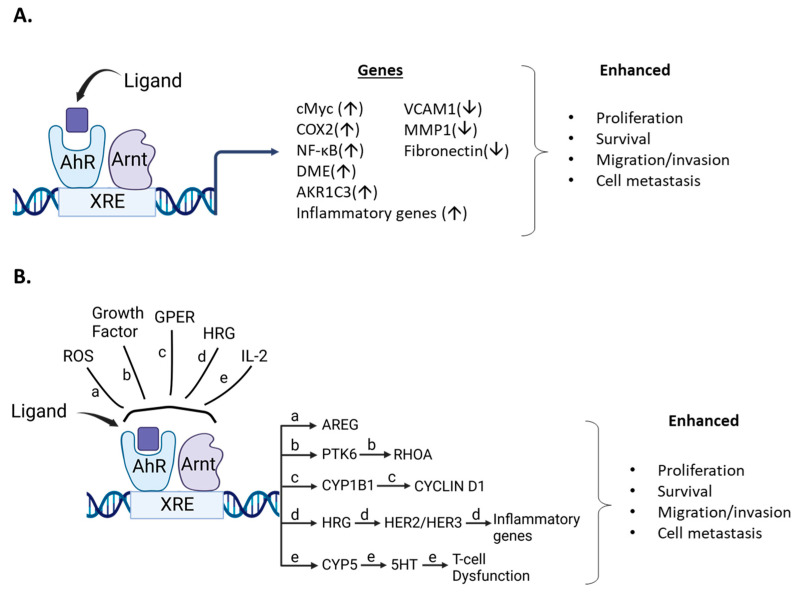
Ligand/AhR-mediated pro-oncogenic activities. (**A**). AhR ligands metabolically activate pro-oncogenic genes/pathways that are inhibited by AhR loss or AhR antagonists. (**B**). Role of AhR in pro-oncogenic pathways that involve other factors resulting in activation of downstream pro-oncogenic pathways.

**Table 1 cancers-14-05574-t001:** TCDD and halogenated aromatics as inhibitors of mammary carcinogenesis.

Compounds	Responses	Cell	KO	In Vivo	Reference
TCDD	multiple	MDA-MB-231 MDA-MB-436 MDA-MB-157 MDA-MB-435 and BT474	✓		[73]
2,3,7,8-TCDF	multiple	MDA-MB-231 MDA-MB-436 MDA-MB-157 MDA-MB-435 and BT474	✓		[73]
2,3,4,7,8-PeCDD	multiple	MDA-MB-231 MDA-MB-436 MDA-MB-157 MDA-MB-435 and BT474	✓		[73]
1,2,3,7,8-PeCDD	multiple	MDA-MB-231 MDA-MB-436 MDA-MB-157 MDA-MB-435 and BT474	✓		[73]
3,3′4,4′,5-PeCB	multiple	MDA-MB-231 MDA-MB-436 MDA-MB-157 MDA-MB-435 and BT474	✓		[73]
TCDD	cell cycle prog.	MCF-7	–	–	[82]
TCDD	gr.	MDA-MB-468	–	✓	[83]
TCDD	gr.	MCF-7	–	–	[84]
TCDD	CXCR4/CXCL12	Multiple			[85]
TCDD	Inv	MDA-MB-231, SKBR3	✓	–	[79]
TCDD	Inv	MDA-MB-231, SKBR3	✓	–	[79]
TCDD	Inv	MDA-MB-231, T47D	✓	–	[86]
TCDD	Inv	MDA-MB-231 and BT474	✓	–	[87]
TCDD	gr	MCF-7	–	✓	[88]
TCDD	met	4T-1	–	✓	[89]

Inv = invasion; met = metastasis; gr = growth.

**Table 2 cancers-14-05574-t002:** Synthetic and pharmaceutical AhR ligands as inhibitors of mammary carcinogenesis.

Compounds	Responses	Cells	KO	In Vivo	Reference
MCDF	gr	MDA-MB-453, MDA-MB-436, HCC-38, MDA-MB-435, BT-474, MDA-MB-157	✓		[73]
MCDF	inv	MDA-MB-231 and BT474	✓	✓	[87]
MCDF	gr	MDA-MB-468	–	✓	[83]
Aminoflavone	gr, DNA damage cytotoxicity, ROS apoptosis	MCF-7, MDA-MB-231, T47D. ZR-75, MDA-MB-468	–	✓	[24,25,26,90,91,92,93,94,95]
Aminobenzothiazoles (DF-203 and SF-203)	gr, ROS, DNA damage	MCF-7, MDA-MB-468, CRL2335, MDA-MB-435	–	✓	[96,97,98,99,100,101,102,103]
Naphthylamide der-invatives (NAP6)	gr	MCF-7, MDA-MB-231, BT26, BT474, MDA-MB-468	–	✓	[104,105]
#12 (quinazoline derivative)	gr, apoptosis, MMP, ROS	MCF-7	–	✓	[106]
ANI -7 (acrylo-nitriles)	gr	MCF-7, T47D, ZR-75 SKBR3, MDA-MB-468, BT20, and BT474	✓	–	[107,108,109]
Aminoglycoside CG3-15943		MDA-MB-468	✓	–	[110,111]
Flutamide	migr.	MDA-MB-468	–	–	[54]
Leflunomide	migr.	MDA-MB-468	–	–	[54]
Nimodipine	migr.	MDA-MB-468	–	–	[54]
Omeprazole	migr.	MDA-MB-468	–	–	[54]
Sulindac	migr.	MDA-MB-468	–	–	[54]
Sulindac	migr	MDA-MB-468	–	–	[54]
Tranilast	inv, migr, gr, met	MDA-MB-468, 4T1	–	✓	[54,113,114,115]
β-Naphthoflavone	gr	MCF-7, MDA-MB-231	✓	–	[116]
Raloxifene	Apoptosis	MDA-MB-231	✓	–	[117]
Carbidopa	multiple	MCF-7, MDA-MB-231		✓	[112]
4-Hydroxytamoxifen	diff	MCF-7	–	–	[118]

gr = growth; inv = invasion; diff = differentiation; migr = migration; met = metastasis; ROS = reactive oxygen species; MMP = mitochondrial membrane potential.

**Table 3 cancers-14-05574-t003:** Endogenous and natural product AhR ligands as inhibitors of mammary carcinogenesis.

Compounds	Responses	Cells	KO	In vivo	Reference
DIM	gr, invasion, met	MDA-MB-231, MCF-7, 4T-1	✓	✓	[78,86,119,120,121,122]
I3C	gr, migr. apoptosis	MIF-7, MDA-MB-231, MDA-MB-468, T47D	–	–	[123,124,125,126]
ICZ	migr	MCF-7, MDA-MB-231,	–	–	[125]
Luteolin	inv, gr, met	MDA-MB-231	–	–	[129]
Icaritin MIR-212/132	gr	MCF-7	✓	✓	[130]
Flavipin	gr, inv, migr	MDA-MB-231, T47D	✓	✓	[131]
Glyceollins CI and II	migr	MDA-MB-231	–	–	[132]
Camalexin	gr, migr (mammosphere)	MCF-7, T47D	–	✓	[133]
2-Hydroxy-6-tridecylbenzone acia	gr	MDA-MB-231	–	–	[134]
Gallic acid	apoptosis, migr, inv, gr	T47D, MDA-MB-231			[135]
ITE		MCF7, MDA-MB-231. MDA-MB-157	✓	–	[136]
FICZ	gr, migr	MCF-7	✓	–	[137]
Indoxylsulfate	ROS, met, migr	4T1	✓	–	[138]
Indolepropionic acid	gr, ROS, met	4T1, SKBR3	✓	–	[139]

gr = growth; migr = migration; inv = invasion; met = metastasis; ROS = reactive oxygen species.

**Table 4 cancers-14-05574-t004:** AhR/AhR ligands enhancing mammary carcinogenesis.

Compound	Responses/Pathway	Cells	KO	In Vivo	Reference
TCDD	gr, Myc/Rel-AhR	Hs5787	–	–	[30,140]
FICZ, BaP, TCDD, XA, Kyn	migr/AhR-TDO-Kyn	SUM149, Hs578T	✓	✓	[141]
CB7993113 DMBA	migr, inv, tox	BP1, Hs5787	✓	–	[142]
FICZ, BaP, TCDD	migr/AhR-SOX2	Hs578T, MCF-7,SUM149	✓	–	[144]
Galangin, NF, MC	gr/genes	Hs5787	✓	–	[143]
DIM, TCDD	colony form, migr	BP1, Hs578T, SUM149, MDA-MB-231	✓	–	[31,145]
Kyn	colonies, inv met/AhR-TDO-KYN, NFkB	BT59, SUM159, MDA-MB-231	✓	–	[146]
BaP	inv, gr, migr	MDA-MB-231	✓	–	[147]
Phthalates	migr, inv/HDAC6	MCF-7, MDA-MB-231	✓	–	[148,149,150]
TCDD, Kyn	surv, infl/COX2, NFkB	MDA-MB-231, SKBR3, others	✓	–	[151]
MC	cytotox/AKR1C3	MDA-MB-231	✓	–	[152]
TCDD	apoptosis, gr, AhRR	MDA-MB-231, MCF-7, others	✓	–	[153]
CH223191	Proangiogenic/AhR-AREG-ROS	multiple	✓	✓	[154]
CH223191	met, migr, motility	MDA-MB-231, Hs578T Others	✓	✓	[155]
MC	gr/AhR-GPER	SKBR3	✓	–	[156]
TCDD, BaP	infl, IL6	MCF-7	✓	–	[157]
MC	migr/HRG-AhR	MCF-7	✓	–	[158]
5-Hydroxtryptophan	IL-2-CD8 +T cell exhaustion	4T1	✓	✓	[159]

migr = migration; gr = growth; inv = invasion; form = formation; surv = survival; infl = inflammation; met = metastasis; ROS = reactive oxygen species.

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
