# Peer review of "The Role of the Aryl Hydrocarbon Receptor (AhR) and Its Ligands in Breast Cancer"

_cancers, 2022, doi:10.3390/cancers14225574_

Round 1

Reviewer 1 Report

This review comprehensively summarizes the research on AhR functions in breast cancer. The authors list both positive and negative aspects of the role of AhR and the ligands in detail. However, in my own view, there are serious problems with the author's description. There seem to be many words and punctuation missing, which makes it tough to understand. The whole article looks like a preliminary draft rather than a scientific paper. The author's work seems to be careless so even the font size of the text is inconsistent.

1) AhR usually expresses in what pathological type of breast cancer? Is it mutually exclusive with the expression of ER?

2) Why do the authors claim that ligands with different structures cause different functions of AhR? Usually, as a receptor, the site of binding to the ligand is constant, and the structure is also constant, so the conformational changes and downstream signals after binding are also relatively constant. There are really few counterexamples. Is there any concrete experimental evidence to support this statement?

3) The authors mention that 4-Hydroxytamoxifen is a ligand of AhR. Which receptor has a higher affinity to 4-Hydroxytamoxifen, ER, or AhR? Is there any SPR or ITC data? It likely binds to ER, so it doesn't work at all in MDA-MB-231. Then a lot of work related to this drug is not reliable.

4) The authors state that "knockdown of the AhR in mice results in abnormalities and lesions in multiple tissues." As a toxicant receptor, which ligand(a toxic compound) works in tissue maintenance? And how normal laboratory mice are exposed to these toxins in a clean environment?

5) The paper cites a number of clinical trials of ligand drugs for AhR. The results themselves are more contradictory, and it is difficult to distinguish precisely whether the efficacy is produced by the drug binding to AhR or to its traditional target.

6) Throughout the article, the authors give a large number of almost completely opposite experimental results, but do not provide a reasonable explanation through proven mechanisms or scientifically supported hypotheses. As in one study cited in the article, TCDD (as the ligand for AhR used primarily in this article) caused growth inhibition in either AhR- or AhR+breast cancer cells, can it be understood that AhR means nothing?

Reviewer 2 Report

This review summarizes the aryl hydrocarbon receptor (AhR)-associated therapies in breast cancer. The topic is fine.

Some typo errors, such as line 34 AhR-dependent and line 95, e.g.: > e.g.,

Line 128, what is ca 50%?

Some of the expression is meaningless, such as ‘Expression of these markers may or may not be indicative of their functional activity’.

In table 1, the meaning of symbols was not listed, and some places lack symbols, which similarly happed in table 2.

In figure 2, there is a lack of names of molecules.

Line 288, ERא?

In figure 4, the expression of only a few genes is listed, what happens for the others?

In figure 5, the results are positive or negative, labeling them is clearer for audiences.

In all the tables, the first line should be highlighted.

Fig 5 should be Figure 5. 

Round 2

Reviewer 1 Report

The most important issue has not been addressed. More than two-thirds of the manuscript is devoted to listing both the cancer-inhibiting and cancer-promoting effects of AhR in breast cancer. But we need a scientific theory, or a well-founded hypothesis, to try to explain the rationality of this difference and to point the way for subsequent studies. Otherwise, the author's statement in the abstract that "future research will be required to resolve these conflicting results" makes no sense. The audiences need a self-justifying theory rather than an aggregation of experimental results.

Reviewer 2 Report

All the comments have been addressed by authors.